# Parechovirus A Pathogenesis and the Enigma of Genotype A-3

**DOI:** 10.3390/v11111062

**Published:** 2019-11-14

**Authors:** Adithya Sridhar, Eveliina Karelehto, Lieke Brouwer, Dasja Pajkrt, Katja C. Wolthers

**Affiliations:** 1Laboratory of Clinical Virology, Department of Medical Microbiology, Amsterdam UMC, location Academic Medical Center, University of Amsterdam, 1100 AZ Amsterdam, The Netherlands; SuviEveliina.Karelehto@ucsf.edu (E.K.); lieke.brouwer@amsterdamumc.nl (L.B.); k.c.wolthers@amsterdamumc.nl (K.C.W.); 2Department of Pediatrics, Emma Children’s Hospital, Amsterdam UMC, location Academic Medical Center, University of Amsterdam, 1100 AZ Amsterdam, The Netherlands; d.pajkrt@amsterdamumc.nl

**Keywords:** parechovirus, parechovirus A, parechovirus A-3, pathogenesis

## Abstract

*Parechovirus A* is a species in the *Parechovirus* genus within the *Picornaviridae* family that can cause severe disease in children. Relatively little is known on *Parechovirus A* epidemiology and pathogenesis. This review aims to explore the *Parechovirus A* literature and highlight the differences between *Parechovirus A* genotypes from a pathogenesis standpoint. In particular, the curious case of Parechovirus-A3 and the genotype-specific disease association will be discussed. Finally, a brief outlook on *Parechovirus A* research is provided.

## 1. Introduction

Human parechoviruses, now officially known as *Parechovirus A* (PeV-A), are common childhood pathogens with a potential for severe clinical manifestations in infants [1,2]. Yet, the current understanding of PeV-A pathogenesis and epidemiology is remarkably limited. While PeV-A shares some similarities to enteroviruses (EVs), research on the biological and clinical aspects of PeV-A infection highlights differences between the genera. Within the PeV-A species, one of the most intriguing open questions is the genotype-specific disease association, i.e., why is severe illness almost exclusively caused by PeV-A3 and not by other genotypes? This review aims to look into this enigma and discuss what is currently known about the prevalence of PeV-A and pathogenesis.

## 2. Genome and Structure

The PeV-A virion is approximately 30 nm in diameter and is composed of a RNA genome shielded by an outer protein capsid lacking a lipid envelope [1,3]. The single-stranded positive-sense (+ss) genome is 7300 nucleotide (nt) long, contains a single open reading frame, and is flanked by untranslated regions (UTRs) [4]. The 700 nt 5′UTR forms primary and secondary RNA structures that are crucial for replication and is attached to a viral protein called VPg [1,4]. An internal ribosomal entry site (IRES) within the 5′UTR facilitates a direct cap-independent translation of a single large polyprotein which is cleaved into three structural proteins (VP0, VP3, and VP1) and seven non-structural proteins (2A–C and 3A–D) [4]. The 300 nt 3′UTR terminates with a polyadenylated tail [4]. 

All picornavirus capsids adopt an icosahedral structure (see Figure 1) [5]. The three capsid proteins assemble into a protomer and five protomers together form a pentamer [5]. A total of 12 pentamers result in the final icosahedron defined by three axes of symmetry: (i) Two-fold axes along the edges of two protomers, (ii) three-fold axes along the protomer triangular faces, and (iii) five-fold axes along the pentamer vertices [6]. However, the atomic structures of PeV-A have revealed several features uncommon among other picornaviruses. The PeV-A capsid surface is relatively flat and misses the classic hydrophobic VP1 pocket (a target for small molecule capsid inhibitors blocking virus uncoating) as is seen in EVs [6,7]. The PeV-A VP0 capsid protein is not cleaved into VP2 and VP4 in the mature virion [4,6]. Furthermore, RNA packaging signals appear to guide the PeV-A capsid assembly [8,9]. Interestingly, procapsids (empty particles devoid of the RNA genome), as seen in many other picornaviruses, are not observed for PeV-A [6,10]. 

## 3. Classification 

*Parechovirus* is a genus within the family *Picornaviridae*, comprising of small non-enveloped, single-stranded positive-sense RNA viruses [11,12]. The genus is comprised of four species, *Parechovirus A*, *Parechovirus B* (formerly named Ljungan virus), *Parechovirus C* (Sebokele virus), and *Parechovirus D* (ferret parechovirus) [13]. Species PeV-A contains virus genotypes that can infect humans and cause severe disease such as meningoencephalitis, seizures, or sepsis-like illness (see Section 6.1. for more information) [14,15]. PeV-A was first isolated as two unidentified viruses in 1956 in the USA from children with diarrhea [16]. They were initially classified as EVs, echovirus 22 and 23, based on their similarity in cytopathogenic effect (CPE), their clinical presentation, and non-pathogenicity in mice and monkeys [16]. In 1999, echovirus 22 and 23 were reclassified as PeV-A1 and PeV-A2, respectively, due to differences in genomic structures, encoded proteins, and other biological properties [4,17,18,19]. In 2004, genotype PeV-A3 was discovered in Japan followed by the discovery of PeV-A4 in the Netherlands in 2006 [20]. Since then the number of PeV-A types increased rapidly with the development of more state-of-the-art molecular techniques. Currently, there are 19 PeV-A types known with PeV-A1 divided into clusters 1A and 1B (Table 1) [21]. 

### Evolution

The PeV-A lineage diverged from its most recent common ancestor around the year 1600 CE, while individual types might have diverged as recently as 150 years ago [34]. As with other picornaviruses, the RNA polymerase of PeV-A lacks proofreading activity and PeV-As have a high mutation rate of approximately 2.5 × 10^−3^ [34]. In addition to this high mutation rate, recombination plays an important part of PeV-A evolution. Recombination has been shown to occur frequently between PeV-A1, A4, A5, and A6, with breakpoints mainly located at both ends of the structural VP1 region, and at the VP2/VP3 junction region [21,35,36]. Recombinant strains with a breakpoint located within the VP1 region are rare [21,35,36]. This is in accordance with the localization of recombination hotspots in other picornaviruses, such as enteroviruses [37].

The location of recombination breakpoints seems to correlate to the genetic diversity in the PeV-A genome, both the nt and amino acid variability are higher in the structural region than in the non-structural region [35]. This illustrates antigenic pressure on the structural region, while purifying selection is dominant in the non-structural region. Recombination might play an important role in establishing this as it would allow for mutations in the structural regions, while detrimental mutations in the non-structural region can be bypassed by adopting a non-structural genome segment from another PeV-A strain. 

## 4. Epidemiology 

There are clear indications that PeV-A circulates worldwide as it was detected in approximately 2% of specimens collected from children with a clinical suspicion of viral infections or as part of the national EV surveillance programs in Japan, Hong Kong, Denmark, Finland, the Netherlands, and the USA [14,15,38,39,40,41,42,43]. Furthermore, a remarkably high prevalence (23–57%) was recently demonstrated for PeV-A in children from Malawi [33,44].

Specific PeV-A genotypic prevalence varies globally. In Europe and the USA, PeV-A1 is most prevalent followed by PeV-A3 and PeV-A4, while PeV-A2, and A7-A19 are rarely reported [38,40,41,42,45,46]. In Asia, similar to Europe and the USA, PeV-A1, A3, and A4 are the most prevalent but a higher diversity of genotypes has been reported in India and Pakistan [47,48,49]. In the African continent, PeV-A1, A2, and A3 are the most prevalent but nearly all PeV genotypes are detected, indicating a much wider circulation of genotypes. Different genotype distributions have further been reported, for example in Pakistan and Ghana, underlining differences in prevalence and diversity between continents [32,50]. These genetic variations arise from not only error-prone RNA-dependent RNA polymerase, but also from recombination (see section on evolution for further details) [35,51]. The frequency of recombination was higher among PeV-A1 and PeV-A4 to A6 sequences as compared to PeV-A3 isolates [35,36]. However, a recent outbreak of neonatal sepsis in Australia was caused by a recombinant PeV-A3 strain [52].

PeV-A infection also shows a clear seasonality with a majority of infections recorded in late summer and fall [53,54,55]. Interestingly, PeV-A3, in contrast to the other genotypes, has been reported to circulate biannually in the Netherlands and Scotland [41,56]. Ultimately, epidemiological studies reporting PeV-A prevalence vary greatly between cohorts in age, clinical presentation, and sample types collected, hampering direct comparisons. Increases in sequencing efforts and standardization of diagnostic PeV-A testing will lead to improved insights on both PeV-A epidemiology and diversity.

In parallel, seroepidemiological studies determine the prevalence of neutralizing antibodies (nAbs) against a specific pathogen in sera of a defined population. Such studies provide a good estimate of virus prevalence, the potential immune status of different patient groups, and circulation in humans. Seropositivity of PeV-A1 nAbs sharply increases in children aged two to five years and is nearly universal in adults based on reports from Finland, the Netherlands, and Japan [57,58,59,60,61]. This is in line with clinical studies showing PeV-A1 as the most common genotype with infections occurring primarily in young children [1]. The seroprevalence of PeV-A2 and PeV-A4 to A6 nAbs is relatively high in Finland, the Netherlands, and Japan [59,61]. Recently, a high overall seroprevalence (68.9%) of PeV-A3 among Dutch, the USA, and Australian populations was shown, which is suggestive of widespread global circulation. Our age-stratified analyses indicated that the infection generally occurs in children younger than 10 years. However, since cross-neutralization among the genotypes PeV-A1/A2 and PeV-A4 to A6 has been observed, unequivocal conclusions about the epidemiology of these genotypes are difficult to base on this data alone [62,63]. 

## 5. Life Cycle 

The PeV-A life cycle after entry is assumed to be similar to other picornaviruses (for review see [64,65]) but experimental evidence is lacking. Briefly, once the virus binds to the host cell surface receptor, endocytosis is initiated. This is followed by uncoating and RNA genome release into the cytoplasm. Then the viral RNA is translated by the host ribosomes into the polyprotein precursor. Subsequently, an extensive remodeling of the intracellular membranes is induced. This results in the formation of either single- or bilayer lipid vesicles termed replication organelles, where replication occurs [66,67,68]. The new RNA can serve as a template for further rounds of replications or be translated in a cap-independent manner into a single large polyprotein. This polyprotein is cleaved into structural and non-structural proteins. Unlike many other picornaviruses, the PeV-A VP0 capsid protein is not cleaved into VP2 and VP4, in the mature virion [4,6]. Another unique feature for PeV-A is that the non-structural 2A protein does not possess proteolytic activity and induce host translation shut-off [19,69]. At the end of the replication cycle, the RNA is encapsidated by the structural proteins to form new infectious particles. The final stage of the PeV-A life cycle is the release from the host cell.

Release can occur through different mechanisms in picornaviruses. Non-enveloped viruses have been traditionally thought to be released via cell lysis. However, several independent reports now point toward an alternative non-lytic egress mechanism. Hepatitis A virus, poliovirus, EV-A71, and coxsackievirus B3 (CV-B3) have all been reported to exit infected cells within host cell derived vesicle structures [70,71,72,73,74]. The proposed mechanism involves hijacking autophagy pathway components to induce the formation of double-membrane vesicles, capable of fusing with the plasma membrane for exit [71,74]. These virus-containing extracellular microvesicles are speculated to facilitate virus dissemination and immune evasion [70,71]. PeV-A is known to release via cell lysis as CPE can be observed in vitro but whether the non-lytic mechanisms are also involved remains to be seen. 

### Receptors 

For most PeV-A genotypes, the receptors involved in viral binding and entry have not been identified and it remains to be seen if there is a universal PeV-A receptor. Presently, nt divergence in the receptor binding capsid VP1-encoding sequence is used to subgroup PeV-A into the 19 detected genotypes [11,17,20]. Some of these genotypes contain an integrin-binding tripeptide arginine-glycine-aspartic acid (RGD) motif in the VP1 C terminus [1,3]. 8 of the 24 known integrin (ITG) heterodimer combinations can bind cellular ligands via the RGD motif [75]. Not surprisingly, other picornaviruses such as echoviruses 1 and 9, and coxsackievirus A9 (CV-A9) have this motif and have been shown to use ITGs as receptors [18,76,77,78]. 

The structural modeling of PeV-A1 indicates that the RGD motif is located between the icosahedral five-fold and three-fold axes on the capsid surface [6,7]. The deletion of the RGD motif results in the loss of PeV-A1 infectivity [79]. Therefore, several groups have investigated the role of αv ITG heterodimers in PeV-A1 infection. Based on a phage display screening, it was suggested that PeV-A1 preferentially uses αvβ1 ITG as a receptor [18]. Experiments employing integrin-blocking monoclonal antibodies (mAbs) indicated αvβ3 ITG as the primary PeV-A1 receptor [80]. One report confirmed that both αvβ3 and αvβ1 integrin heterodimers were involved in PeV-A1 infection while others describe αvβ6 ITG as the high affinity receptor for PeV-A1 [81]. More recently, it was reported that PeV-A1 infection occurs via αVβ1 ITG [82]. Taken together, these results indicate that integrins facilitate PeV-A1 infection but the use of specific heterodimers may be cell line dependent. While ITGs are important for virus binding and entry in cell lines, ITGs may not be exclusive receptors in vivo. In studies on human airway epithelial (HAE) cultures, we found no effect of αv ITG blocking on PeV-A1 and PeV-A3 infection [83]. Interestingly, PeV-A1 strains lacking VP1 C-terminal RGD sequence have been isolated from a Dutch child [84]. Furthermore, PeV-A3 and PeV-A7 to A19 lack the RGD motif and ITGs are assumed not to be involved in entry [1,25]. However, the lack of the RGD motif does not preclude ITG binding and integrin-ligand interactions can occur in a RGD-independent manner. For example, the very late antigen 2 (also known as α2β1 ITG) has been shown to mediate echovirus 1 infection in this way [77,85,86].

The presence of a RGD sequence does not exclude the involvement of other entry receptors and ITG independent infection has been described in RGD motif containing viruses such as CV-A9 [87]. We showed that heparin sulfate (HS) is involved in PeV-A1 infection in the human lung adenocarcinoma A549 cell line [88]. However, we could not confirm the role of HS in the HAE model [83]. In this model, we also observed that both PeV-A1 (with RGD in VP1) and PeV-A3 (RGD-less) preferentially infect basal cells from the basolateral surface (see Figure 2) [83]. Either, these closely related viruses are utilizing separate receptors to gain entry into the same cell type or perhaps they share a yet unidentified receptor. If it is the latter, then one interesting possibility is beta-2-microglobulin (β2M), a component of the antigen-presenting major histocompatibility complex I (MHC 1) [89,90]. Initially, β2M was shown not to be involved in the PeV-A1 infection of A549 and RD cell lines [79,80]. However, recent reports show β2M blocking significantly reduced PeV-A1 infection in the SW480 cell line and co-localization of β2M and PeV-A1 was observed in the A549 cell line [87]. 

As a side note, β2M binding draws interesting parallels to echoviruses, under which the PeV-A was initially classified. Some echoviruses such as echovirus 1 use α2β1 ITG as a primary receptor but this ITG is not a receptor for other echoviruses [77]. Recently, the neonatal Fc receptor (FcRn) was shown to be a pan-echovirus receptor [91]. FcRn has a binding partner in β2M and loss of β2M expression renders cells resistant to echovirus infection. Perhaps, through β2M, PeV-A share more than clinical similarities to echoviruses.

In conclusion, the exact receptors used by PeV-A remains an open question but ITGs, HS, and β2M play a role in viral binding. The identification of receptors in physiologically relevant models for PeV-A will be a step forward in understanding parechovirus pathogenesis. 

## 6. Pathogenesis 

Due to the lack of suitable model systems, the current understanding of PeV-A pathogenesis is very limited. Thus, the primary replication site and initial cell tropism as well as target organs (secondary infection sites) are unknown. PeV-A infection is presumed to occur via fecal-oral and respiratory routes based on similarities to infection routes of EVs and PeV-A detection in stool and nasopharyngeal (NP) samples [92]. While there is no direct experimental evidence to support this hypothesis, transmission via the fecal-oral route is likely based on the prolonged stool shedding of this virus [93,94]. 

### 6.1. Clinical Manifestation and Detection 

Clinical manifestations following PeV-A infection range from asymptomatic or mild symptoms to severe disease. While a definitive association with symptomatology has not been established for many of the PeV-A genotypes, the causal relationship of PeV-A3 infections with a number of clinical conditions is evident [95]. PeV-A1 is associated with acute gastroenteritis, upper respiratory tract symptoms, fever, and rash in children between 6 months to 5 years [14,15,45]. In addition to these rather mild symptoms, PeV-A3 often causes severe disease in infants under the age of 3 months [1,3]. Several outbreaks of neonatal sepsis-like illness due to PeV-A3 infection have also been recorded, most recently in Australia [96,97,98,99,100]. Central nervous system (CNS) syndromes such as acute flaccid paralysis, meningitis, and encephalitis have been reported for PeV-A1 and A3 [2,4,25,101]. However, CNS complications are more commonly associated with PeV-A3 infection and in some instances, long-term neurodevelopmental sequelae have been reported [101,102,103,104,105]. It must be noted that these long-term neurological complications are rare and further studies are needed to formally assess long-term outcomes [106,107,108]. Epidemics of myalgia and a case of myocarditis in PeV-A3 infected adults have been reported but overall, PeV-A detection in adults is rare [1,109,110,111,112]. 

PeV-As can be detected in feces, NP swabs, blood, and cerebrospinal fluid (CSF) [1]. Most PeV-A epidemiological studies are performed using feces samples from children younger than 5 years of age, presenting with symptoms of acute gastroenteritis [38,45,47,113,114]. Furthermore, some published reports focus on children with respiratory or neurological symptoms with NP, CSF, and serum samples [2,115,116,117,118,119]. Diagnostic detection of PeV-A is performed by RT-qPCR targeting the untranslated region (UTR) of the 5′ end which is conserved among all PeV-A genotypes [92,120]. The PeV-A genotype is determined by the partial or complete sequencing of the VP-1 encoding region but genotyping is not routinely performed [92]. PeV-A1-A6 can be detected by virus culture, but PeV-A3 needs different cell lines than the other PeVs and compared to molecular detection, this method is less sensitive [92].

### 6.2. Immune Response 

Once the virus enters the body, the host immune responses are engaged to actively counter the viral infection. Unfortunately, the information on host immune responses against PeV-A is scarce. In general, the host innate and adaptive immune response appear to efficiently control the PeV-A spread as infections are largely self-resolving. Pattern recognition receptors, such as toll-like receptors (TLRs), melanoma differentiation-associated gene 5, and retinoic acid-inducible gene I, form key host defense mechanisms against intracellular pathogens [5,121]. In case of PeV-A1, TLR7, and TLR8 are important in recognizing the single stranded RNA early in the infection [122]. This recognition subsequently activates the NF-κβ signaling pathway and ultimately leads to the secretion of proinflammatory cytokines such as interferon β (IFN-β), tumor necrosis factor a (TNF-α), and interleukin 6 (IL-6). It has been shown that PeV-A1 infection induces the type 1 interferon response by phosphorylation of the upstream IFN regulatory factor 3 [123]. More recently, PeV-A1 was also shown to activate the signaling downstream of TLR3 [124]. How the PeV-A1 proteins modulate these pathways is not known. We investigated the genotypic differences in the innate immune response between PeV-A1 and PeV-A3 using an adult HAE model system. We observed that PeV-A3 induced a significantly stronger activation of genes involved in immune and inflammatory signaling [83]. This initial host response may dictate the viral spread and different clinical outcomes between PeV-A1 and PeV-A3.

The role of T-cell immunity in picornavirus infection is not well-defined. In a single study published on T-cell immunity against PeV-A infection, reactivity to the capsid proteins was observed in 20 adults by measuring T-cell proliferation upon exposure to recombinant VP0, VP3, and VP1 proteins [125]. Increased T-cell proliferation in response to PeV-A1 capsids indicate that T-cell immunity could play a role during PeV-A infection. 

With respect to the antibody responses, although the administration of intravenous immunoglobulins (IVIG) in an immunodeficient adolescent with a chronic PeV-A1 infection did not reduce the PeV-A1 viral load [126], IVIG administration was associated with the full recovery of a PeV-A1 infection in a child presenting with severe dilated cardiomyopathy [127]. In animal studies, PeV-A1 VP1 C terminus has been shown to be immunogenic. Rabbits immunized with peptide antigen corresponding to the VP1 region elicited nAbs against PeV-A1 [57]. In terms of the neutralizing antibody responses, the RGD motif appears to play a role. A human monoclonal antibody epitope was mapped to the VP1 C-terminal RGD-region of PeV-A1 [128]. nAbs targeting the RGD motif cross neutralize other PeV-A genotypes (PeV-A2 and PeV-A4 to A6) that contain this motif [57,62]. Other immunodominant epitopes include the N-terminal region of the PeV-A1 VP0 capsid encoding region and VP0/VP3 loops on the capsid surface [128]. 

During the first 3 to 6 months of life, maternal nAbs protect neonates against infection. While maternal nAbs protect against disease related to most PeV-A genotypes, PeV-A3 is again an exception. Severely ill neonates lacked maternal nAbs against PeV-A3 at the onset of a disease and mothers from neonates with PeV-A3 infections showed no or low a-PeV-A3 Ab titers [129,130]. 

## 7. PeV-A3 Stands Out Amongst the PeV-A Genotypes 

Although the review has focused on the PeV-A species as a whole, special attention is warranted for PeV-A3 which is repeatedly observed as an exception amongst the PeV-A genotypes. For instance, the lack of a RGD motif and biannual circulation of PeV-A3 in comparison to other PeV-A genotypes are interesting. Furthermore, PeV-A3 rarely recombines with any of the other types but recombination does occur within the genotype [131]. The most striking feature of PeV-A3 infection is the severity in its clinical manifestation. The reason behind the severe clinical presentation of PeV-A3 as opposed to the more prevalent PeV-A1 remains a critical question. The clinical differences in PeV-A1 vs PeV-A3 could perhaps be linked to the immune response to these genotypes. 

### Host Responses Gone Astray? 

Both the innate and adaptive immune responses have been shown to be different for PeV-A3 as compared to the other PeV-As. Our group showed on HAE cultures, PeV-A3 induced a much stronger activation of genes involved in immune and inflammatory signaling [83]. In neonates with an immature or still developing immune system, this strong activation could result in severe immunopathology. This is supported by the fact that elevated levels of proinflammatory cytokines and systemic inflammatory response have been linked to the severe pathogenesis of EV-A71 [132,133,134] while the concept of cytokine storm is well established for respiratory pathogens such as influenza virus and human coronavirus [135,136].

Neonatal sepsis, also known as systemic inflammatory response syndrome, is the most common severe clinical manifestation of PeV-A3 and points towards immunopathology during the PeV-A3 infection of neonates. Neonatal sepsis is associated with the increased vascular permeability and reduced blood-brain barrier (BBB) function. Viremia and increased BBB permeability resulting from neonatal sepsis during PeV-A3 infection may allow passive virus leakage into the CNS [137,138,139,140]. Thus, detection of PeV-A3 RNA in the CSF samples of severely ill infants may not be a sign of active virus replication in the CNS but rather a consequence of a systemic inflammatory response. This is supported by several studies reporting that PeV-A3 positive CSF samples often do not show increase in white blood cell count [96,141,142,143,144,145]. However, as a counter to this, our group has shown that PeV-A3 is capable of infecting neuronal cell lines and PeV-A3 isolates replicated human neuroblastoma cell lines more efficiently than PeV-A1, suggesting a neural tropism of PeV-A3 [146]. Therefore, viral replication in the CNS could potentially lead to encephalitis, meningitis, and adverse long-term neurodevelopmental outcomes that is seen in pediatric PeV-A3 infection. 

Another reason for exacerbated disease in neonates upon PeV-A3 infection might be related to humoral immunity. We and others have shown the titers of PeV-A3 nAbs are below the expected levels of protection in adults over 30 years [61,130,147]. This is further supported by the lack of neutralization of PeV-A3 in a small serosurvey of adults from Wisconsin, USA [148]. This is different from PeV-A1 where the nAb levels are maintained in adults. No cross protection from PeV-A1 antibodies is expected for PeV-A3 and it has been shown that hyperimmune serum raised in rabbits against PeV-A1 did not cross-neutralize PeV-A3 [62]. Thus, low nAb titers in women of childbearing age may lead to inadequate maternal antibody protection contributing to PeV-A3 outbreaks in infants. Hence, we speculate that the severe PeV-A3 presentation in young infants is due to both insufficient (adaptive) and excessive (innate) host immune responses.

## 8. Treatment 

Currently, there are no specific antiviral therapies clinically available against PeV-A infection. Treatment options are limited to supportive care and occasionally passive immunization by the administration of IVIG [3]. The development of targeted antiviral drugs requires detailed knowledge of the virus lifecycle but this is poorly defined [149]. Few studies have screened EV inhibitors against PeV-A but these have indicated that the capsid morphology, protease functions, and host factor dependency differ between these two genera [149]. For instance, pleconaril (an EV capsid inhibitor) was unable to inhibit PeV-A replication due to differences in the capsid structures, whereby large amino acid side chains block the hydrophobic drug binding site on the surface of PeV-A [6,7,126]. Similarly, EV 3C protease, EV 2C inhibitors, and golgi transport inhibitor (Brefeldin A, effective against EV) failed to inhibit PeV-A1 replication [68,150,151]. Another host factor-targeting compound, the antifungal agent Itraconazole (ITZ), was shown to have antiviral activity against PeV-A3 but not against PeV-A1 [152]. ITZ is highly interesting as it is a FDA-approved drug and has a favorable safety profile in pediatric patients. Thus, ITZ looks promising for PeV-A3 treatment but further research into the pathogenesis of PeV-A is needed to design targeted antiviral therapies. 

Given that maternal nAbs appear to be protective against severe disease related to PeV-A infection, the development of therapeutic mAbs is an unmet clinical need [62]. We have isolated and characterized the structure and function of human PeV-A mAbs in detail and found that several of them neutralize PeV-A1 and other RGD-containing genotypes efficiently [128]. One of the mAbs is capable of neutralizing PeV-A3 but unfortunately only the antigenically distinct prototype strain and not the circulating clinical strains [6,153,154]. Nevertheless, these data represent a promising step towards antibody treatment of PeV-A infections.

## 9. Outlook 

In this review, we discussed the epidemiology of PeV-A and summarized the current understanding in PeV-A pathogenesis. However, information on the prevalence and pathogenesis of PeV-A is limited. PeV-A studies have relied on the use of immortalized cell lines that, while useful, are still poor models for pathogenesis studies. Animal models expressing virus-specific human receptors have been the gold standard in picornavirus research but no such model exists for PeV-A infection. Newborn mice and cynomolgus monkeys have been experimentally infected with PeV-A1 and newborn mice with PeV-A3 but with no apparent pathologies associated [16,25]. In order to elucidate entry and pathogenesis, the identification of a PeV-A receptor is critical and may lead to the generation of a transgenic animal model. 

Alternatively, the use of complex organotypic human cell culture models could be a way forward in understanding PeV-A pathogenesis. One such model is the well-differentiated HAE cell culture system in which primary human epithelial cells from airway tissue are cultured on transwell inserts. Our group used this model for studying PeV-A infection and identified basal cells as the cell type that PeV-A1 and A3 infect in the airway. A recent breakthrough in similarly culturing the gut epithelium using stem cells from intestinal crypts could also be used for studying PeV-A [155]. These primary culture systems, termed organoids, have been developed for various organs including the brain [156]. They recapitulate the cell composition, organization, and tissue microenvironment in vitro and will be valuable for understanding pathogenesis in the human setting. 

Another important reason to expand our understanding of PeV-A pathogenesis is to address the lack of therapeutic options. Currently, there is no rationale for vaccine development but efforts such as the antigenic profiling of PeV-A genotypes is crucial and may lead to the discovery of therapeutic monoclonal antibodies. Alternatively, knowledge of the PeV-A replication cycle could result in the development of targeted antiviral compounds.

Perhaps, one of the most intriguing open question in PeV-A biology is the genotype specific disease association. Why is severe illness almost exclusively caused by PeV-A3 and not by other genotypes? We have hypothesized based on experimental findings in HAE models and observed humoral response that this could be due to the immune response against PeV-A3 infection. Further experiments are needed to assess whether this is indeed the case. Future research in this area should also aim to characterize the molecular and cellular determinants of PeV-A immune modulation across various tissues in both infants and adults.

In conclusion, PeV-A are common childhood pathogens with a potential for severe clinical disease in infants. While current knowledge is limited, it presents a unique opportunity for picornavirologists to explore a clinically relevant pathogen. We anticipate that identification of a receptor and use of organotypic models will drive the PeV-A research field forward in the coming years. 

## Figures and Tables

**Figure 1 viruses-11-01062-f001:**
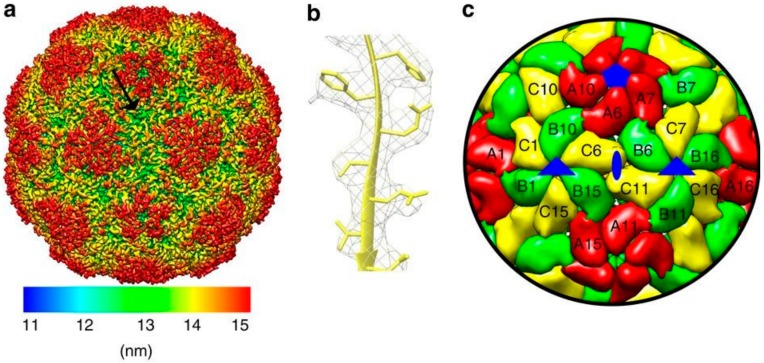
(**a**) Twofold axis of symmetry of PeV-A3 (*Parechovirus A*) at 4.3 Å resolution, canyon indicated by the arrow, (**b**) representative fit of the VP0 atomic model, and (**c**) capsid model showing positions of VP0 (yellow), VP1 (red), and VP3 (green). In blue are the symmetry axes—fivefold pentagon, threefold triangle, and twofold ellipse. Image used under the terms of the Creative Commons Attribution License (CC BY 4.0.) from Shakeel, S. et al., 2016 [6].

**Figure 2 viruses-11-01062-f002:**
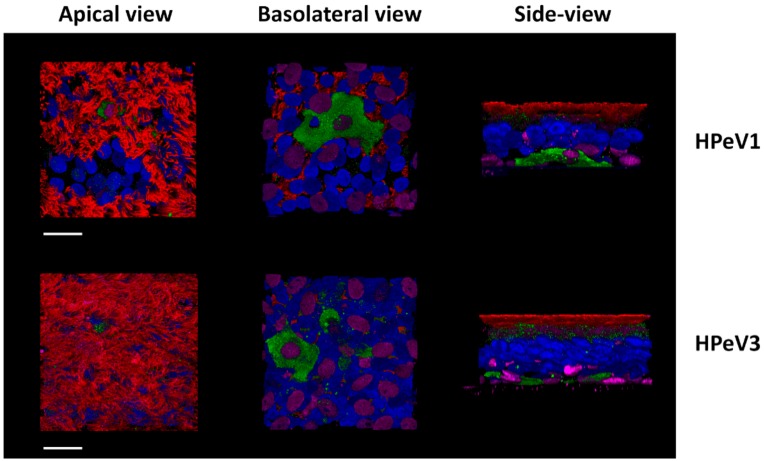
Immunofluorescence image of human airway epithelia (HAE) inserts infected with PeV-A1 and PeV-A3. Both genotypes appear to have a tropism for basal cells in the airway epithelium. Legend: Blue—DAPI (nuclei), Red—β-tub (ciliated cells), Green—PeV-A, and purple—p63 (basal cells). Image used under the terms of the Creative Commons Attribution License (CC BY 4.0.) from Karelehto E. et al., 2018 [83].

**Table 1 viruses-11-01062-t001:** (Left) Select list of PeV-A prototype strains (http://www.picornastudygroup.com/) [13]. Full list of prototype strains available on the picorna study group website.

Type	Strain	Reference	Accession
PeV-A1A	Harris	Hyypia et al., 1992 [22]	L02971
PeV-A1B	BNI-788 St	Baumgarte et al., 2008 [23]	EF051629
PeV-A2	Williamson	Ghazi et al., 1998 [24]	AJ005695
PeV-A3	A308/99	Ito et al., 2004 [25]	AB084913
PeV-A4	K251176-02	Benschop et al., 2006b [20]	DQ315670
PeV-A5	CT86-6760	Oberste et al., 1998 [17]	AF055846
PeV-A6	NII561-2000	Watanabe et al., 2007 [26]	AB252582
PeV-A7	PAK5045	Li et al., 2009 [27]	EU556224
PeV-A8	BR/217/2006	Drexler et al., 2009 [28]	EU716175
PeV-A9	BAN2004-10902	Nix et al., 2013 [29]	JX219575
PeV-A10	BAN2004-10903	Nix et al., 2013 [29]	JX219568
PeV-A11	BAN2004-10905	Nix et al., 2013 [29]	JX219574
PeV-A12	BAN2004-10904	Nix et al., 2013 [29]	JX219567
PeV-A13	BAN2004-10901	Nix et al., 2013 [29]	JX219579
PeV-A14	451564	Benschop et al., 2008c [30]	FJ373179
PeV-A15	BAN-11614	Nix et al., 2013 [29]	JX219573
PeV-A16	BAN-11615	Nix et al., 2013 [29]	JX219580
PeV-A17	M36/CI/2014	Böttcher et al., 2017 [31]	KT319121
PeV-A18	GhanaA36 886	Graul et al., 2017 [32]	KY931660
PeV-A19	P02-4058	Brouwer et al., 2019 [33]	MH339678

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
