# Peer review of "Parechovirus A Pathogenesis and the Enigma of Genotype A-3"

_viruses, 2019, doi:10.3390/v11111062_

Round 1
Reviewer 1 Report
The present review of Sridhar, Karelehto et al. covers in detail and sensible manner the epidemiological findings and pathogenesis of Human Parechoviruses, with the main emphasis in PeV-A serotypes. Also, revise its basic biology, usually characteristics shared with other picornaviruses. Moreover, explains differences between PeV-A serotypes in the pathogenesis of this group of viruses. The review very informative and add light into this group of human picornaviruses. In the opinion of the reviewer, the article is a bit convoluted and could profit with some rewriting, aiming to deliver concisely the messages to reach the audience. Especially it is advised to polish the “brief outlook on Parechovirus A research.” A few concerns are raised in the comment section.
Main observations:
The maximum likelihood tree of the VP1 sequences is in itself a result, and in the opinion of this reviewer, a review is not the place to report results because there is not a method section that clearly explains how the results were generated. Moreover, in the accompanying table 1, some citations are missing in the reference section, as Böttcher et al., 2017. Besides, the claimed PeV-A19 have the reference Böttcher et al., unpub. It is recommended to refer to the article and published tables in Brouwer L, Karelehto E, Han AX, et al. High frequency and diversity of parechovirus A in a cohort of Malawian children. Arch Virol. 2019;164(3):799–806. doi:10.1007/s00705-018-04131-7
A careful reorganization of the sections of the article it is advised, to facilitate the reading and comprehension. I.e., the subsections “5.1. Genome and structure”, “5.2. Receptors” and “5.3. Entry, life cycle, and release” should not be comprised inside the section “5. Pathogenesis”, since there is no much discussion about how those elements affect the PeV-A serotypes pathogenesis distinctively. Moreover, the title subsection “5.3. Entry, life cycle, and release” (line 191) is in part misleading, in the opinion of the reviewer. Entry and release are most usually described as part of the virus life cycle, not a separate part. Consider reformulate this title. Besides, the lengthy discussion of in the subsection “5.2. Receptors”, should be considered the start point of the virus cycle, since initiates with attachment. The mentioned subsection is recognized in the next subsection since “The PeV-A life cycle after entry is assumed to be similar to other picornaviruses […], but experimental evidence is lacking. Briefly, once the virus binds to the host cell surface receptor...” (line 192).
The paragraph “5. Pathogenesis” could be followed with “5.4. Immune response”
In line 202, the sentence ”At the end of the replication cycle, the RNA is encapsidated by the structural proteins to form new infectious particles in a process called morphogenesis.” Since it is a specialist journal, the reviewer thinks morphogenesis should not be defined, since virus morphogenesis is a broadly used term by the virology-specialist audience.
In line 195, “...cytoplasm. Subsequently,...”, it looks there are missing events to comprehend the process, maybe “...cytoplasm. [Then the viral RNA is translated by the host ribosomes into the polyprotein precursor….] Subsequently,...”. The infectious particle does not carry with the 3Dpol and other necessary non-structural proteins for replication. For the reader to deepen on these matters, it is suggested to cite :Jiang, P., Liu, Y., Ma, H.-C., Paul, A. V & Wimmer, E. Picornavirus morphogenesis. Microbiol. Mol. Biol. Rev. 78, 418–37 (2014).
In line 38 “...were initially classified as EVs, echovirus 22 and 23, based on their similarity in cytopathogenic effect...” is not clearly consonant with the discussion in lines 212-215. Furthermore, since this article is a review, in the opinion of this reviewer, it is not part of the contemporary standard to refer to personal observations and unpublished data, in lines 212-215.
A reduced version of the subsection “5.5. Evolution” may fit better as introductory paragraph before in the paragraph of classification. This change would result in giving continuity between “5.4. Immune response” and “PeV-A3 stands out amongst the PeV-A genotypes”, which could be fuse with the next section by removing the subsection title “6.1. Host responses gone astray?”.
In line 319, “For instance, pleconaril – an EV capsid inhibitor was unable to inhibit PeV-A replication likely due to the unique flat features of the PeV-A structure”, it is known the plecoranil binds into the pocket in VP1, so the flatness of the virus do not compromise it. Is it the pocket closed or narrower?
Author Response
We would like to thank the reviewer for taking the time to review our manuscript and highly appreciate the detailed feedback provided. We have addressed their specific points below:
Reviewer comment: The present review of Sridhar, Karelehto et al. covers in detail and sensible manner the epidemiological findings and pathogenesis of Human Parechoviruses, with the main emphasis in PeV-A serotypes. Also, revise its basic biology, usually characteristics shared with other picornaviruses. Moreover, explains differences between PeV-A serotypes in the pathogenesis of this group of viruses. The review very informative and add light into this group of human picornaviruses. In the opinion of the reviewer, the article is a bit convoluted and could profit with some rewriting, aiming to deliver concisely the messages to reach the audience. Especially it is advised to polish the “brief outlook on Parechovirus A research.” A few concerns are raised in the comment section.
Our response: We have made minor changes to section 9. Outlook. We have written this as an opinion piece on where PeV-A research could head in the future. We couldn’t pinpoint which aspect needs polishing but if the reviewer were to provide specific feedback on this, we are happy to revise.
Reviewer comment: The maximum likelihood tree of the VP1 sequences is in itself a result, and in the opinion of this reviewer, a review is not the place to report results because there is not a method section that clearly explains how the results were generated.
Our response: We agree that the maximum likelihood tree is in itself a result and given that the method of how the tree was generated is out of the scope of this review, we have excluded the tree from the manuscript.
Reviewer comment: Moreover, in the accompanying table 1, some citations are missing in the reference section, as Böttcher et al., 2017. Besides, the claimed PeV-A19 have the reference Böttcher et al., unpub. It is recommended to refer to the article and published tables in Brouwer L, Karelehto E, Han AX, et al. High frequency and diversity of parechovirus A in a cohort of Malawian children. Arch Virol. 2019;164(3):799–806. doi:10.1007/s00705-018-04131-7.
Our response: Table 1 has been modified to include the references and accession numbers. Initially, we included Bottcher et al., unpublished as the strain for PeV-A19 as this was listed by the picornastudy group. However, the study group website has since been amended with the reference strain (Brouwer et al., 2019). We have now included that reference for PeV-A19.
Reviewer comment 3: A careful reorganization of the sections of the article it is advised, to facilitate the reading and comprehension. I.e., the subsections “5.1. Genome and structure”, “5.2. Receptors” and “5.3. Entry, life cycle, and release” should not be comprised inside the section “5. Pathogenesis”, since there is no much discussion about how those elements affect the PeV-A serotypes pathogenesis distinctively.
Our response: We have reorganized the sections of the article as per the reviewer’s suggestions. 5.1, 5.2, and 5.3 in the first version of the manuscript have been removed from the pathogenesis section. Genome and structure have been moved to earlier in the manuscript and is now section 2. Entry, life cycle, and release has been made into a separate section (5. Life cycle) with the receptors as a subsection within this section.
Reviewers comment: Moreover, the title subsection “5.3. Entry, life cycle, and release” (line 191) is in part misleading, in the opinion of the reviewer. Entry and release are most usually described as part of the virus life cycle, not a separate part. Consider reformulate this title. Besides, the lengthy discussion of in the subsection “5.2. Receptors”, should be considered the start point of the virus cycle, since initiates with attachment. The mentioned subsection is recognized in the next subsection since “The PeV-A life cycle after entry is assumed to be similar to other picornaviruses […], but experimental evidence is lacking. Briefly, once the virus binds to the host cell surface receptor...” (line 192).
Our response: The title of this section has been shortened to “5. Life cycle”. While we agree with the reviewer that the receptor section should be considered as starting point of the virus life cycle, we have for the sake of clarity kept the receptor work under a separate heading. Receptor has been made into a subsection (5.1. Receptors) within the life cycle section.
Reviewers comment: The paragraph “5. Pathogenesis” could be followed with “5.4. Immune response”
Our response: This has been adjusted accordingly.
Reviewers comment: In line 202, the sentence ”At the end of the replication cycle, the RNA is encapsidated by the structural proteins to form new infectious particles in a process called morphogenesis.” Since it is a specialist journal, the reviewer thinks morphogenesis should not be defined, since virus morphogenesis is a broadly used term by the virology-specialist audience.
Our response: The text “in a process called morphogenesis” has been removed from line 136-137.
Reviewers comment: In line 195, “...cytoplasm. Subsequently,...”, it looks there are missing events to comprehend the process, maybe “...cytoplasm. [Then the viral RNA is translated by the host ribosomes into the polyprotein precursor….] Subsequently,...”. The infectious particle does not carry with the 3Dpol and other necessary non-structural proteins for replication. For the reader to deepen on these matters, it is suggested to cite :Jiang, P., Liu, Y., Ma, H.-C., Paul, A. V & Wimmer, E. Picornavirus morphogenesis. Microbiol. Mol. Biol. Rev. 78, 418–37 (2014).
Our response: We have added the line “Then the viral RNA is translated by the host ribosomes into the polyprotein precursor” in line 128. We have also included the suggest review as a recommended reading in the start of the section in line 126.
Reviewers comment: In line 38 “...were initially classified as EVs, echovirus 22 and 23, based on their similarity in cytopathogenic effect...” is not clearly consonant with the discussion in lines 212-215. Furthermore, since this article is a review, in the opinion of this reviewer, it is not part of the contemporary standard to refer to personal observations and unpublished data, in lines 212-215.
Our response: We have removed the personal observations and unpublished data. Instead, that text has been replaced with “PeV-A is known to release via cell lysis as CPE can be observed in vitro but whether the non-lytic mechanisms are also involved remains to be seen.” in section 5, line 146-147.
Reviewers comment: A reduced version of the subsection “5.5. Evolution” may fit better as introductory paragraph before in the paragraph of classification. This change would result in giving continuity between “5.4. Immune response” and “PeV-A3 stands out amongst the PeV-A genotypes”, which could be fuse with the next section by removing the subsection title “6.1. Host responses gone astray?”.
Our response: The evolution section has been moved into the classification section. Now the “PeV-A3 stands out amongst the PeV-A genotypes” follow the “Immune response” section. However, we have kept the “PeV-A3 stands out amongst the PeV-A genotypes” as a separate section instead of making it into a subsection in “Pathogenesis. We believe that having it as a separate section helps emphasize the focus on PeV-A3 that is alluded in the title.
Reviewers comment: In line 319, “For instance, pleconaril – an EV capsid inhibitor was unable to inhibit PeV-A replication likely due to the unique flat features of the PeV-A structure”, it is known the plecoranil binds into the pocket in VP1, so the flatness of the virus do not compromise it. Is it the pocket closed or narrower?
Our response: Pleconaril is unable to inhibit PeV-A replication as amino acid side chains block the drug binding site on the surface. We have changed the sentence (line 314-316) to “……. PeV-A replication due to differences in the capsid structures; large amino acid side chains block the hydrophobic drug binding site on the surface of PeV-A”. Two references for this have been added.
Reviewer 2 Report
Although the pathogenesis aspects are well-summarised (though still very speculative), the clinical aspects are rather biased and skewed - mainly due to the references this team has cited.
Part of the problem is that I suspect that these authors are lab-based and not clinicians.
You don't need an animal model to confirm transmission via the faecal-oral route in human infections - many have no good animal model and indeed animal models may give misleading information.
Saitoh and colleagues have shown that the faecal-oral route is a very likely and plausible transmission route for PeV - which makes sense, virologically, given the prolonged stool shedding of this virus:
https://www.ncbi.nlm.nih.gov/pubmed/30184210
https://www.ncbi.nlm.nih.gov/pubmed/26305831
And other studies from the UK have not demonstrated frequent, major, long-term neurological sequelae in PeV A-3 infections - and that such long-term complications are relatively rare:
https://www.ncbi.nlm.nih.gov/pubmed/30001232
https://www.ncbi.nlm.nih.gov/pubmed/29871901
https://www.ncbi.nlm.nih.gov/pubmed/30530486
https://www.ncbi.nlm.nih.gov/pubmed/23347531
The authors need to provide a more balance literature review and not omit some significant studies from outside their collaborative circle.
Author Response
We would like to thank the reviewer for taking the time to review our manuscript and highly appreciate the detailed feedback provided. We are in agreement with the comments from reviewer 2 but feel that the phrasing in the previous version of the manuscript does not reflect that. We have adjusted the manuscript appropriately.
Reviewers comment: Although the pathogenesis aspects are well-summarised (though still very speculative), the clinical aspects are rather biased and skewed - mainly due to the references this team has cited. Part of the problem is that I suspect that these authors are lab-based and not clinicians. You don't need an animal model to confirm transmission via the faecal-oral route in human infections - many have no good animal model and indeed animal models may give misleading information.
Our response: We did not intend to imply that an animal model is necessary for confirming transmission via the fecal-oral route. Rather, that statement was mainly to highlight that the understanding of PeV-A pathogenesis is very limited. However, the sentence structure leads to that interpretation and have adjusted this section accordingly. The introduction to pathogenesis (section 6, line 197) now reads “Due to the lack of suitable model systems, the current understanding of PeV-A pathogenesis is very limited. Thus, the primary replication site and initial cell tropism as well as target organs (secondary infection sites) are unknown. PeV-A infection is presumed to occur via fecal-oral and respiratory routes based on similarities to infection routes of EVs and PeV-A detection in stool and NP samples.”
Reviewers comment: Saitoh and colleagues have shown that the faecal-oral route is a very likely and plausible transmission route for PeV - which makes sense, virologically, given the prolonged stool shedding of this virus:
https://www.ncbi.nlm.nih.gov/pubmed/30184210
https://www.ncbi.nlm.nih.gov/pubmed/26305831
Our response: We agree that fecal-oral route is very likely and a plausible transmission route based on clinical observations. We have added the sentence in line 201 “While there is no direct experimental evidence to support this hypothesis, transmission via the fecal-oral route is likely based on prolonged stool shedding of this virus.” to reflect this. We have also included those two references to that statement.
Reviewers comment: And other studies from the UK have not demonstrated frequent, major, long-term neurological sequelae in PeV A-3 infections - and that such long-term complications are relatively rare:
https://www.ncbi.nlm.nih.gov/pubmed/30001232
https://www.ncbi.nlm.nih.gov/pubmed/29871901
https://www.ncbi.nlm.nih.gov/pubmed/30530486
https://www.ncbi.nlm.nih.gov/pubmed/23347531
The authors need to provide a more balance literature review and not omit some significant studies from outside their collaborative circle.
Our response: We did not intend to imply that long-term neurological complications from PeV-A3 infections are common but rather that they can occur. We have amended the sentence in line 213 to include “…. in some instances, long-term neurodevelopmental sequelae have been reported”. This also corresponds to the references provided by the reviewer. Furthermore, in order to clarify that the neurological complications are rare, the following sentence “It must be noted that these long-term neurological complications are rare and further studies are needed to formally assess long-term outcomes” has been added in line 216. The references provided by the reviewer have been cited here.
Round 2
Reviewer 2 Report
A detailed, comprehensive review - improved now with the rephrasing of some sentences and the inclusion of additional relevant papers.